# Role of the Sodium-Dependent Organic Anion Transporter (SOAT/SLC10A6) in Physiology and Pathophysiology

**DOI:** 10.3390/ijms24129926

**Published:** 2023-06-08

**Authors:** Marie Wannowius, Emre Karakus, Zekeriya Aktürk, Janina Breuer, Joachim Geyer

**Affiliations:** 1Institute of Pharmacology and Toxicology, Faculty of Veterinary Medicine, Biomedical Research Center Seltersberg (BFS), Justus Liebig University of Giessen, Schubertstr. 81, 35392 Giessen, Germany; marie.t.wannowius@vetmed.uni-giessen.de (M.W.); emre.karakus@vetmed.uni-giessen.de (E.K.); janina.breuer@vetmed.uni-giessen.de (J.B.); 2General Practice, Faculty of Medicine, University of Augsburg, 86159 Augsburg, Germany; zekeriya.akturk@gmail.com

**Keywords:** SOAT, *SLC10A6*, sulfated steroids, transport, breast cancer, inhibitor

## Abstract

The sodium-dependent organic anion transporter (SOAT, gene symbol *SLC10A6*) specifically transports 3′- and 17′-monosulfated steroid hormones, such as estrone sulfate and dehydroepiandrosterone sulfate, into specific target cells. These biologically inactive sulfo-conjugated steroids occur in high concentrations in the blood circulation and serve as precursors for the intracrine formation of active estrogens and androgens that contribute to the overall regulation of steroids in many peripheral tissues. Although SOAT expression has been detected in several hormone-responsive peripheral tissues, its quantitative contribution to steroid sulfate uptake in different organs is still not completely clear. Given this fact, the present review provides a comprehensive overview of the current knowledge about the SOAT by summarizing all experimental findings obtained since its first cloning in 2004 and by processing SOAT/*SLC10A6*-related data from genome-wide protein and mRNA expression databases. In conclusion, despite a significantly increased understanding of the function and physiological significance of the SOAT over the past 20 years, further studies are needed to finally establish it as a potential drug target for endocrine-based therapy of steroid-responsive diseases such as hormone-dependent breast cancer.

## 1. Introduction

Over the last two decades, the sodium-dependent organic anion transporter (SOAT, gene symbol *SLC10A6*) has been well-established as a specific carrier for sulfated steroid hormones in humans, rats, and mice [1,2,3,4]. The SOAT is a transmembrane protein with nine transmembrane domains (TMDs) that acts as a sodium-dependent uptake carrier for compounds such as dehydroepiandrosterone sulfate (DHEAS), estrone sulfate (E1S), or pregnenolone sulfate (PREGS) [2]. These sulfo-conjugated, biologically inactive steroid metabolites occur in quite high concentrations in the blood circulation and serve as precursors for the intracrine formation of active estrogens and androgens that contribute to the overall regulation of steroids in many peripheral tissues [5,6]. As sulfated steroids are negatively charged at physiological pH, uptake carriers such as SOAT are required to import these molecules for subsequent intracellular cleavage of the sulfate group by the steroid sulfatase STS (so-called sulfatase pathway) [6]. Based on this, only target cells that express uptake carriers such as the SOAT, together with STS, and estrogen/androgen receptors can respond to sulfated steroid hormones [7]. Although SOAT expression has been detected in several peripheral tissues, its quantitative contribution to the steroid sulfate uptake and sulfatase pathway in different organs is still not completely clear. Furthermore, the role of the SOAT in disease progression, particularly in steroid-responsive tumors, and its pharmaceutical potential, e.g., for anti-proliferative therapy, require further investigation. Given these facts, the present review provides a comprehensive overview of the current knowledge about the SOAT by summarizing all experimental findings obtained since its first cloning in 2004 and by processing SOAT/*SLC10A6*-related data from genome-wide protein and mRNA expression databases.

## 2. The SLC10 Carrier Family

The SOAT is assigned to the solute carrier family 10 (SLC10) based on its high phylogenetic relationship to the other six members of this carrier family [1,8,9]. The SLC10 family was established in the early 1990s when the first two members, the Na^+^/taurocholate co-transporting polypeptide (NTCP, gene symbol *SLC10A1*) and the apical sodium-dependent bile acid transporter (ASBT, gene symbol *SLC10A2*), were identified by expression cloning [10,11]. The NTCP and ASBT are essentially involved in the maintenance of the enterohepatic circulation of bile acids (Figure 1) by mediating sodium-dependent bile acid uptake in the liver and intestine, respectively [9,12]. Consistent with this function, the SLC10 family was formerly referred to as the “sodium/bile acid co-transporter family” [8]. Later, in the early 2000s, five more genes, referred to as *SLC10A3-SLC10A7*, were identified that phylogenetically belong to this carrier family [9,12]. Among them, *Slc10a6*/Soat was first cloned from rat adrenal gland mRNA in 2004 [2]. Just a few years later, the human *SLC10A6*/SOAT homolog was cloned and functionally characterized [1]. Interestingly, the SOAT showed no transport activity for the most physiologically important bile acids such as taurocholic acid and glycocholic acid, which are the typical substrates of the NTCP and ASBT (Table 1). Instead, the SOAT revealed specific transport activity for all physiologically relevant 3′- and 17′-monosulfated steroid hormones [1,3,13,14]. In addition, only two minor bile acid species, namely taurolithocholic acid-3-sulfate (TLCS) and taurolithocholic acid (TLC), were found to be transported by the SOAT [1,14].

Regarding the other four members of the SLC10 carrier family (namely SLC10A3, SLC10A4, SLC10A5, and SLC10A7), transport experiments in transfected HEK293 cells or *Xenopus laevis* oocytes revealed no transport activity for bile acids or sulfated steroid hormones at all. Therefore, these carriers are still classified as orphan carriers [9,15]. The SLC10A4 protein was detected in the synaptic vesicles of cholinergic and monoaminergic neurons in the central and peripheral nervous system [16,17]. Although no direct transport activity of the SLC10A4 protein could be detected for neurotransmitters such as dopamine, norepinephrine, serotonin, histamine, aspartate, GABA, glutamate, ATP, and acetylcholine [18], SLC10A4 appears to be involved in the regulation of the vesicular accumulation of dopamine and acetylcholine, at least in mice [19,20]. *SLC10A5* showed the highest mRNA expression levels in the liver and kidney [21], and the phenotypic characterization of a corresponding knockout mouse model is ongoing in our laboratory. *SLC10A7* was first cloned in 2007 and represents the most distant member of the SLC10 carrier family, with a low sequence homology of only <15% to the NTCP, ASBT, and SOAT [22] (Table 1). Patients with mutations in the *SLC10A7* gene presented skeletal dysplasia with multiple large joint dislocations, short stature, and amelogenesis imperfecta [23,24,25]. More recently, the SLC10A7 protein has been renamed as “negative regulator of intracellular calcium signaling” (RCAS) due to its significant effect on store-operated calcium entry (SOCE) via the plasma membrane [26,27]. Taken together, our current knowledge of the SLC10 carrier family clearly points to functions beyond bile acid transport [9,15].

## 3. Genomic Organization of the *SLC10A6/Slc10a6* Genes

Phylogenetically, the SOAT and ASBT are most closely related and share identical gene structures. All *SLC10A6/Slc10a6* and *SLC10A2/Slc10a2* (uppercase for humans, lowercase for animals) genes identified to date have six coding exons with highly conserved exon/intron boundaries, suggesting that both of them share a common ancestor gene [9]. In contrast, the human *SLC10A1*, *SLC10A4*, *SLC10A5*, and *SLC10A7* genes exhibit different gene structures with five, three, one, and 11/12 coding exons, respectively. The open reading frame of the human *SLC10A6* transcript consists of 1134 base pairs (Figure 2A) and codes for the 377-amino-acid (aa) SOAT protein [1] (Figure 2B). The Soat proteins in rats and mice are slightly shorter, exhibiting 370 and 373 aa, respectively [2,4].

## 4. SOAT Protein Structure, Sorting, and Dimerization

Based on the crystal structures of two bacterial SLC10-homologous proteins from *Neisseria meningitidis* and *Yersinia frederiksenii* named ASBT_Nm_ and ASBT_Yf_, respectively [28,29], and the more recent cryogenic electron microscopy (cryo-EM) structures of human NTCP [30,31,32,33], the SOAT is suggested to have a nine-TMD structure with N_exo_/C_cyt_ orientation (Figure 2B). This membrane topology is supported by the de novo structure prediction of the SOAT with AlphaFold [34]. Within this structure, TMDs 1, 5, and 6 form a panel domain, and the other TMDs (2/3/4 and 7/8/9) form a core domain. Within the core domain, TMDs 2/3/4 and 7/8/9 are topologically similar but oppositely orientated within the membrane, revealing an internal twofold pseudosymmetry. TMDs 3 and 8 are discontinuous and cross each other, thereby facilitating the binding of two Na^+^ ions as the co-substrates (Figure 2C). Substrate binding and translocation occur at the interface between both domains [32].

On the protein level, the SOAT shows the highest sequence homology with the ASBT (42% sequence identity, 70% sequence similarity), followed by the NTCP (33% sequence identity, 63% sequence similarity) [8,9]. The SOAT is a 46 kDa glycoprotein when expressed in HEK293 cells. After PNGase treatment, the apparent molecular weight was found to drop to 42 kDa, most likely representing the non-glycosylated SOAT core protein [1]. There are three potential N-glycosylation sites for the SOAT at amino acid positions Asn^4^, Asn^14^, and Asn^157^, but it is not yet clear where the glycosylation of the SOAT protein exactly occurs (Figure 2B). Sorting studies in transfected HEK293 cells clearly localized the SOAT protein to the plasma membrane [1,13,35,36]. However, immunohistochemistry (IHC) analysis of SOAT expression in different tissues also showed large parts of the protein in the Golgi compartment, at least in primary spermatocytes [13].

Recently, it was shown that homo- and heterodimerization is a common feature of all SLC10 carriers. Different experimental approaches, such as Western blot, co-immunoprecipitation, chemical cross-linking, and membrane-based yeast-two-hybrid screening, have clearly demonstrated the presence of homodimers for the SOAT, NTCP, and ASBT [37,38,39,40,41]. For all three proteins, dimerization seems to be relevant for their regulation and proper expression at the plasma membrane. Interestingly, heterodimerization between different SLC10 carriers has also been demonstrated (e.g., for NTCP/SOAT, NTCP/ASBT, ASBT/SOAT, or SLC10A4/SOAT), suggesting the presence of highly conserved dimerization domains in all SLC10 proteins [41].

## 5. SOAT Substrate Docking and Proposed Transport Mechanism

Very recently, cryo-EM structures of the human NTCP protein were independently solved by four research groups. These studies used detergent-solubilized recombinant NTCP or recombinant NTCP reconstituted in nanodiscs for high-resolution cryo-EM [30,31,32,33]. Essentially, three different conformations of the NTCP protein have been identified: inward-open conformation (PDB 7pqg [31]), outward-open conformation (PDB 7wsi [30], PDB 7fci [33], and PDB 7vad [30]), and open-pore conformation (PDB 7pqq [31] and PDB 7zyi [32]). Due to the high sequence homology between the NTCP and the SOAT, it is assumed that there are also similar protein conformations for the SOAT (Figure 3).

Based on these NTCP structures, different kinds of transport mechanisms have been proposed. Transitions between the outward-open and inward-open conformations could facilitate an alternating access transport mechanism, where the substrate-binding site has alternating access to the extra- and intracellular milieu and the binding of the co-substrate Na^+^ is important for inducing the conformational transition [29,30]. Alternatively, for the open-pore conformation of the NTCP, it was proposed that one bile acid molecule binds to an outer substrate-binding site (S_out_), thereby preventing ion and water leakage by sealing the open pore. This bile acid molecule can then shift to a second binding site (S_in_) while the transporter is reloading an additional substrate to S_out_. The substrate then can be released from S_in_ to the intracellular milieu while the substrate binding at S_out_ is still sealing the open pore [32]. Finally, it cannot be excluded that the open-pore conformation of the NTCP is just an intermediate state of the transport cycle [31] so the transporter may undergo all three resolved conformations during the transport cycle.

In order to illustrate substrate binding and translocation for the SOAT, homology models were generated for representative structures of all three conformations, namely outward-open (Figure 3A), open-pore (Figure 3B), and inward-open (Figure 3C). These structures were then used for the virtual docking of DHEAS, the prototypic SOAT substrate. In the outward-open conformation, DHEAS binds at the bottom of a large outward accessible cavity, without allowing translocation of the substrate to the intracellular milieu in this conformation. In contrast, in the inward-open conformation, this entrance cavity is structurally covered at the top, and the DHEAS molecule has free access to the intracellular milieu. Based on this, the transition of the SOAT protein from the outward-open to the inward-open conformation may be sufficient to translocate the DHEAS molecule from the outside to the inside of the cell. As the DHEAS transport via the SOAT is strictly sodium-dependent [1], it is very likely that two Na^+^ ions act as co-substrates, thereby contributing to the conformational transition of the SOAT protein during the transport cycle, as demonstrated for the bacterial ASBT and human NTCP proteins [28,29,30,31,32,33]. However, based on the docking data for the open-pore conformation, it also seems possible that the protein traverses this conformation as an intermediate state. This would involve rotating the DHEAS molecule and positioning it in an orientation with the sulfate group toward the top (Figure 3B), from which it can be released in the inward-open conformation (Figure 3C). During this proposed transport cycle, the core domain seems to move in a slight lateral movement away from the panel domain, thereby allowing DHEAS to pass through the protein.

However, it must be emphasized that these data were only generated on homology models of the human SOAT and the actual structural conformations of the SOAT must be determined experimentally. Direct structural comparisons between the NTCP and the SOAT would be highly valuable for clarifying the differences in the ligand-binding and substrate translocation behavior of both proteins (NTCP and SOAT) at the molecular level. Although both carriers bind and translocate DHEAS as a substrate, only the NTCP accepts bile acids as additional substrates, whereas bile acids only bind to the SOAT without being translocated [1].

## 6. SOAT Substrate Recognition

The functional characterization of the SOAT has mostly been carried out in stably transfected Flp-In T-REx HEK293 cells, where the SOAT cDNA was stably integrated at the Flp recognition target (FRT) site, and SOAT expression could be induced by tetracycline treatment [1,3]. Systematic transport studies were performed using different classes of steroids through (I) radiolabeled substrates and liquid scintillation counting (e.g., for [^3^H]DHEAS, [^3^H]E1S, [^3^H]TLC), or (II) liquid chromatography-tandem mass spectrometry (LC-MS/MS) detection of unlabeled compounds (e.g., for 17β-estradiol-3,17-disulfate, 17α-OH-pregnenolone sulfate, testosterone sulfate) [1,3]. In these studies, the SOAT proved to be a very specific carrier for all physiological 3′- and 17′-monosulfated steroid hormones (see Table 2). In contrast, non-sulfated steroids (e.g., estrone, DHEA), glucuronidated steroids (e.g., estradiol-17β-D-glucuronide, estrone-3β-D-glucuronide), and disulfated steroids such as 17β-estradiol-3,17-disulfate were not transported [1,3,14]. This indicates that the presence of one single negatively charged sulfate group at the steroid nucleus is required for the SOAT substrates (Figure 4A). Thereby, α- or β-orientations of the sulfate group were accepted at both the 3′ and 17′ positions, as indicated by the substrate pairs androsterone sulfate (3α) and epiandrosterone sulfate (3β), and epitestosterone sulfate (17α) and testosterone sulfate (17β) (Table 2) [1,3,14].

All of these 3′- and 17′-monosulfated steroids are characterized by a lipophilic steroid backbone with a planar structure (A/B-*trans*, B/C-*trans*, C/D-*trans*) and a terminal negatively charged sulfate group [48]. The substrate recognition of both substrate groups can be explained by the pseudo-symmetry of the 3′- and 17′-monosulfated steroids, which show a close structural overlap when rotated by 180° against each other. Based on this, both 3′- and 17′-monosulfated steroids likely bind to the same substrate-binding site but in different orientations [3]. However, a second sulfate group might cause it to structurally exceed the substrate-binding pocket, thereby preventing the transportation of disulfated steroids such as 17β-estradiol-3,17-disulfate by the SOAT. In contrast, other modifications at the steroid backbone, such as 5α-reduction, 16α-hydroxylation, and 17α-hydroxylation were found to be acceptable for substrate recognition. Therefore, the compound pairs 5α-dihydrotestosterone sulfate and testosterone sulfate, 16α-OH-dehydroepiandrosterone sulfate and DHEAS, and 17α-OH-pregnenolone sulfate and pregnenolone sulfate could all be transported by the SOAT [3].

**Table 2 ijms-24-09926-t002:** SOAT substrates.

Substrates	Substrate K_m_
**Steroid 3’-monosulfates**	
Pregnenolone sulfate (PREGS)	11.3 µM [1,3]
Estrone sulfate (E1S)	12.0 µM [1,13]
Dehydroepiandrosterone sulfate (DHEAS)	28.7 µM [1]
16α-OH-DHEAS	319.0 µM [49]
17α-OH-PREGS	n.d. [3]
Androstenediol-3-sulfate	n.d. [13]
Androsterone sulfate	n.d. [3]
Epiandrosterone sulfate	n.d. [3]
17β-estradiol-3-sulfate	n.d. [13]
**Steroid 17’-monosulfates**	
17β-estradiol-17-sulfate	n.d. [3]
5α-dihydrotestosterone sulfate	n.d. [3]
Epitestosterone sulfate	n.d. [3]
Testosterone sulfate	n.d. [3]
**Bile acids**	
Taurolithocholic acid	19.3 µM [14]
Taurolithocholic acid-3-sulfate	n.d. [1]
**Non-steroidal organosulfates**	
2-Sulfooxymethylpyrene (2-SMP)	n.d. [1]
4-Sulfooxymethylpyrene (4-SMP)	n.d. [1]

Note: n.d. = K_m_ not determined.

The 3D planar tetracyclic ring structure of the steroid hormones with the typical A/B-*trans*, B/C-*trans*, and C/D-*trans* conformation seems to be important for substrate recognition by the SOAT. This may explain why the planar organosulfates 2-sulfooxymethylpyrene (2-SMP) and 4-sulfooxymethylpyrene (4-SMP) are also recognized as substrates. However, steroid-based structures that deviate from this planar structure are generally not recognized as SOAT substrates. This applies to almost all bile acids (e.g., taurocholic acid, glycocholic acid) with the typical A/B-*cis*, B/C-*trans*, and C/D-*trans* conformation of the steroid ring system, as well as heart glycosides such as digoxin and ouabain with an A/B-*cis*, B/C-*trans*, and C/D-*cis* conformation. There are only two exceptions to this rule, namely the sulfated bile acid TLCS [1] and the secondary taurine-conjugated bile acid TLC [14], both of which are transported by the SOAT. Interestingly, TLC is the only substrate identified so far that is common to the NTCP, ASBT, and SOAT, suggesting that it binds to a highly conserved substrate-binding site among these three carriers that may be specific to this substrate and could explain this peculiarity in the substrate recognition of TLC and TLCS.

## 7. Sodium-Dependent Transport Mode of SOAT

The NTCP and ASBT are well-established sodium-dependent co-transporters for Na^+^ and bile acids [9,12,15]. In addition, the SOAT mediates the sodium-coupled co-transport of its substrates, and when sodium chloride in the transport buffer is replaced by an equimolar concentration of choline chloride, SOAT transport is completely abolished [1,2]. An electrogenic BA: Na^+^ transport stoichiometry of 2:1 was found for the NTCP and ASBT, which means that two positively charged Na^+^ ions are transported together with one negatively charged bile acid molecule [8]. In addition, the ASBT and NTCP structures mentioned above clearly demonstrate two distinct Na^+^ binding sites within the core domain of these proteins, clearly supporting this transport mode and substrate stoichiometry [28,32]. Interestingly, all amino acids involved in coordinative Na^+^ binding are highly conserved between the NTCP, ASBT, and SOAT. The Na^+^ binding site 1 (Na1) is formed by S105/S112/S112, N106/N113/N113, S119/S126/S126, T123/T130/T130, and E257/E261/E261, and the second Na^+^ binding site 2 (Na2) binds sodium via Q68/Q75/Q75, E257/E261/E261, T258/T262/T262, C260/M264/A264, and Q261/Q265/Q265 (amino acid positions in NTCP/ASBT/SOAT, respectively). However, there is one difference between the NTCP, ASBT, and SOAT regarding the acceptance of other monovalent cations as co-substrates. When sodium chloride in the transport buffer was replaced by equimolar concentrations of lithium chloride, bile acid transport via the NTCP and ASBT was completely abrogated, whereas the SOAT retained ~40% residual transport activity for DHEAS, even with Li^+^ as the co-substrate [1]. This indicates that although the substrate recognition of the SOAT seems to be much more restricted than that of the NTCP and ASBT, the SOAT is less selective in the acceptance of monovalent cations as co-substrates.

## 8. SOAT Genetic Variants

Genetic variants, including the more common single-nucleotide polymorphisms (SNPs) and rare genetic variants, in carrier genes often result in disease-associated transporter dysfunction or loss-of-function phenotypes [50]. In the case of the SOAT, several hundred naturally occurring genetic variants have been identified through the systematic screening of different SNP databases. Among them, 30 non-synonymous variants were analyzed more closely, all of which were predicted to affect the transport function [35,36]. All variants were analyzed for plasma membrane sorting and DHEAS transport function. Six variants, namely A83V, P107L, L204F, G241D, G263E, and Y308N, showed significantly lower membrane expression, leading to reduced transport activity. Seven other variants were completely transport deficient, even when they were properly sorted to the plasma membrane, namely L44P, Q75R, G109S, S112F, N113K, S133F, and G294R [35,36]. Although no clear genotype–phenotype correlation could be identified for the SOAT L204F variant in subjects with hypospermatogenesis (see Section 12.1), in the future, these loss-of-function genetic variants could potentially be used to subgroup patient populations based not only on high and low SOAT expression levels (see Section 12.1) but also on genotype-based functional and dysfunctional SOAT variants.

## 9. SOAT Inhibitors

The SOAT has been established as a potential drug target for anti-proliferative therapy of breast cancer cells (see Section 12.3) [51]. In addition, the inhibition of SOAT-mediated import of sulfated steroids using pharmacological inhibitors may be beneficial for other steroid-responsive tissues such as the skin, endometrium, adipose tissue, or prostate [52]. Therefore, structure–activity relationship (SAR) analyses of SOAT inhibitors are of particular interest (Table 3 and Figure 4B). First, it must be noted that some of the physiological bile acids, although not transported by the SOAT, are quite good SOAT inhibitors [1]. The SOAT inhibition potencies of monomeric bile acids have been ranked in the following order: TLCS (IC_50_ = 0.5 µM) > TLC (IC_50_ = 3.0–3.9 µM) > lithocholic acid 3-sulfate (IC_50_ = 4.2 µM) > lithocholic acid (IC_50_ = 10.4 µM) > chenodeoxycholic acid (IC_50_ = 11.2 µM) > glycochenodeoxycholic acid (IC_50_ = 26.6 µM) > taurochenodeoxycholic acid (IC_50_ = 38.1 µM) > glycodeoxycholic acid (IC_50_ = 46.7 µM) > tauroursodeoxycholic acid (IC_50_ = 49.6 µM) [48]. Of note, the dimeric bile acid molecules S 0960 (IC_50_ = 0.15 µM), S 1690 (IC_50_ = 0.7 µM), and S 3068 (IC_50_ = 1 µM) were found to be even more potent SOAT inhibitors as monomeric bile acid molecules [48]. However, these dimeric bile acids do not have any physiological relevance. In addition, some other plant-derived steroid-like molecules have shown potent SOAT inhibition, e.g., the heart glycoside digitonin (IC_50_ = 4.1 µM), betulinic acid (IC_50_ = 1.2 µM), or the betulin-derivatives SAL-II-68 (IC_50_ = 3.6 µM) and EMe I 4 (IC_50_ = 5.4 µM) [14,48] (Table 3).

Systematic screening of SOAT inhibition was also performed using a set of xenobiotic non-steroidal organosulfates. Here, 1-(ω-sulfooxyethyl)pyrene (1ω-SEP), bromosulfophthalein (BSP), 2-SMP, and 4-SMP were all found to be potent SOAT inhibitors [1]. Subsequent studies on BSP revealed an IC_50_ value of 3.6 µM [48], whereas 2-SMP and 4-SMP were identified as SOAT substrates (see Table 2). In contrast, other sulfo-conjugated organic molecules, such as ethylsulfate, phenylsulfate, phenylethylsulfate, 2-propylsulfate, 5-sulfooxymethylfurfural, hydroquinone sulfate, 4-methylumbelliferylsulfate, and indoxylsulfate [1], had little or no inhibitory effect on the SOAT transport of DHEAS (Appendix A). Furthermore, among a series of differently substituted alpha-naphthyl derivatives, only alpha-naphthylsulfate showed SOAT inhibition, whereas alpha-naphthylisothiocyanate, alpha-naphthylphosphate, and alpha-naphtylamine did not show inhibitory effects, indicating that a sulfate group is an essential structural requirement for small organic SOAT inhibitors [1].

Apart from these bile acids and organosulfates, some additional molecules showed strong inhibition of the SOAT transporter, namely the phenyl sulfonamide S 1647 (IC_50_ = 1.1 µM); benzothiazepine S 0382 (IC_50_ = 75.2 µM); barbiturate derivative S 3740 (IC_50_ = 1.1 µM); propanolamines S 8214 (IC_50_ = 14.9 µM), S 9202 (IC_50_ = 18.7 µM), S 9086 (IC_50_ = 23.1 µM), S 9203 (IC_50_ = 37.7 µM), and S 9087 (IC_50_ = 50 µM); as well as the drugs troglitazone, irbesartan, losartan, and cyclosporine A [14,48] (Figure 4B and Table 3).

**Table 3 ijms-24-09926-t003:** SOAT inhibitors of different chemical classes.

Inhibitor	Inhibitor IC_50_
**Monomeric and dimeric bile acids**	
S 0960 (dimeric)	0.15 µM [48]
Taurolithocholic acid-3-sulfate (TLCS) *	0.5 µM [48]
S 1690 (dimeric)	0.7 µM [48]
S 3068 (dimeric)	1.0 µM [48]
Taurolithocholic acid (TLC) *	3.0–3.9 µM [14,48]
Lithocholic acid-3-sulfate	4.2 µM [48]
Lithocholic acid	10.4 µM [48]
Chenodeoxycholic acid	11.2 µM [48]
Glycochenodeoxycholic acid	26.6 µM [48]
Taurochenodeoxycholic acid	38.1 µM [48]
Glycodeoxycholic acid	46.7 µM [48]
Glycolithocholic acid-3-sulfate	46.7 µM [48]
Tauroursodeoxycholic acid	49.6 µM [48]
Taurodeoxycholic acid	76.3 µM [48]
Taurocholic acid	65.3–99.7 µM [14,48]
Deoxycholic acid	100.1 µM [48]
Glycoursodeoxycholic acid	100.8 µM [48]
7-Ketolithocholic acid	164.2 µM [48]
Hyodeoxycholic acid	172.6 µM [48]
Cholic acid	177.6 µM [48]
Glycocholic acid	284.8 µM [48]
Ursododeoxycholic acid	384.8 µM [48]
Hyocholic acid	971.8 µM [48]
Glycolithocholic acid	n.d. [1]
**Steroids and steroid sulfates**	
Digitonin	4.1 µM [48]
Pregnenolone-3-sulfate (PREGS) *	9.1 µM [48]
Estrone-3-sulfate (E1S) *	22.1 µM [48]
RR Scymnol sulfate	23.3 µM [48]
Cortisone	29.6 µM [48]
Estriol	~100 µM [48]
17β-estradiol-3,17-disulfate	133.2 µM [48]
17β-estradiol-3-sulfate *	145.9 µM [48]
Corticosterone-21-sulfate	323.8 µM [48]
**Betulin derivatives**	
Betulinic acid	1.2 µM [14]
SAL-II-68	3.6 µM [48]
EMe I 4	5.4 µM [48]
SAL-II-156	66.9 µM [48]
3-O-Caffeoyl betulin	301.1 µM [14]
Lupenone	664.5 µM [14]
Betulin	912.2 µM [14]
**Non-steroidal organosulfates**	
Bromosulfophthalein (BSP)	3.6 µM [14,48]
4-Methylumbelliferyl sulfate	255.7 µM [48]
1-(ω-sulfooxyethyl)pyrene	n.d. [1]
2-Sulfooxymethylpyrene (2-SMP) *	n.d. [1]
4-Sulfooxymethylpyrene (4-SMP) *	n.d. [1]
α-Naphthyl sulfate	n.d. [1]
**Others**	
S 1647	1.1 µM [48]
S 3740	1.1 µM [48]
T 5854015	9.0 µM [48]
T 0511-1698	15.0 µM [48]
L-Thyroxine	49.5 µM [48]
T 5573915	57.0 µM [48]
S 0382	75.2 µM [48]
T 5239532	137.0 µM [48]
Cyclosporine A	n.d. [14]
Erythrosine B	n.d. [14]
Ginkgolic acid 17:1	n.d. [14]
Troglitazone	n.d. [14]
Irbesartan	n.d. [14]
Losartan	n.d. [14]
**Propanolamine derivatives**	
S 8214	14.9 µM [48]
S 9202	18.7 µM [48]
S 9086	23.1 µM [48]
S 9203	37.7 µM [48]
S 9087	50.0 µM [48]

Note: n.d. = not determined; * substrate inhibitor.

Using SAR analysis of 25 structurally divergent SOAT inhibitors that covered four orders of magnitude of IC_50_ values, a 3D pharmacophore model for the SOAT was established. The best hypothesis for the calculated SOAT pharmacophore model consisted of three hydrophobic sites and two hydrogen bond acceptors. This model achieved good prediction values and was used to screen a library of ~12 million commercially available compounds. Among the identified hits, four SOAT inhibitors with distinct chemical structures were validated experimentally. All of them demonstrated SOAT inhibition, although with varying potencies of IC_50_ = 9 µM (T 5854015), 15 µM (T 0511-1698), 57 µM (T 5573915), and 137 µM (T 5239532), respectively [48].

In addition to their partly overlapping and partly complementary substrate patterns, the NTCP, ASBT, and SOAT shared some inhibitors but also showed certain carrier inhibition specificities. A systematic comparison of an inhibitor set of 14 chemically divergent molecules indicated no significant correlation of the IC_50_ values between the most closely related carriers ASBT and SOAT. However, certain individual compounds were found to be quite good inhibitors against both carriers, e.g., S 3740 (IC_50_ = 8 µM for ASBT and 1.1 µM for SOAT), S 1647 (IC_50_ = 4 µM for ASBT and 1 µM for SOAT), and S 3068 (IC_50_ = 4 µM for ASBT and 1 µM for SOAT), clearly indicating some overlap in the inhibitor-binding sites of the two carriers. In another study, the inhibitory potencies of four different betulin derivatives were comparatively analyzed for the NTCP, ASBT, and SOAT. This study showed that there was no common inhibition pattern observed for either the individual carriers or the betulin derivative used as the inhibitor. As an example, betulinic acid was found to be a potent inhibitor for the NTCP and SOAT, with an IC_50_ value of ~1 µM but did not show any inhibition of the ASBT at all. In contrast, troglitazone and BSP were identified as potent inhibitors of all three carriers and, therefore, can be regarded as pan-SLC10 inhibitors [14]. However, it must be noted that all inhibitors of the SOAT at least partially interacted with one of the bile acid carriers so a SOAT-specific inhibitor could not be identified so far.

## 10. SOAT Tissue Expression Pattern

The first screening of the SOAT expression pattern was performed in 2007 by real-time PCR quantification of the *SLC10A6* mRNA expression in a cDNA tissue panel comprising 16 major human organs [1]. Later, a larger human tissue cDNA panel comprising 24 organs was analyzed [13]. Both analyses revealed the highest *SLC10A6* mRNA expression in the testis, and high expression levels in the skin, vagina, kidney, pancreas, placenta, mammary gland, lung, and heart. The high expression of the *SLC10A6* mRNA transcript in the testis was further verified through RT-PCR analysis of human testicular biopsies [13]. Recently, more comprehensive SOAT protein and *SLC10A6* mRNA expression data have become available through the Human Protein Atlas (HPA, www.proteinatlas.org), which provides genome-wide expression data obtained through systematic proteomic and transcriptomic analyses. The SOAT protein and *SLC10A6* mRNA expression data provided by the HPA are classified with an enhanced reliability score, which represents the highest level of reliability offered by the HPA. The expression data provided by the HPA are presented in Figure 5, Figure 6, Figure 9 and Figure 11. In addition, *SLC10A6* mRNA expression data from the Genotype-Tissue Expression (GTEx) project (www.gtexportal.org (accessed on 29–31 March 2022 and on 20 June 2022)) were used. The GTEx is another database that provides genome-wide gene expression data. The GTEx data are shown in Figure 5 and Figure 7. 

In Figure 5A, the HPA and GTEx *SLC10A6* mRNA expression data are presented for 58 different human organs. The highest *SLC10A6* mRNA expression levels were detected in the skin, esophagus, adipose tissue, vagina, testis, breast, cervix, and salivary gland. In addition, Figure 5A indicates all organs in which SOAT protein expression was detected by IHC, including the breast, bronchus, cervix, esophagus, nasopharynx, oral mucosa, prostate, skin, stomach, tonsil, and vagina. Figure 5B,C illustrate SOAT expression in females and males, respectively.

A more detailed analysis of the IHC images revealed a very specific and typical expression pattern for the SOAT, which was mainly localized to epithelial structures in different tissues (Figure 6). Here, the basal cell layer of all epithelial structures and the glandular cells in secretory structures showed intense SOAT immunoreactivity. Accordingly, high SOAT expression was detected in the basal cell layer of the bronchial epithelium, cervix epithelium, esophagus epithelium, nasopharyngeal epithelium, oral mucosa, skin, and vaginal epithelium. In addition, strong SOAT staining was detected in the ductal epithelium of the breast, glandular epithelium of the prostate, gastric gland epithelium of the stomach, and squamous epithelial cells of the tonsil (Figure 6).

**Figure 5 ijms-24-09926-f005:**
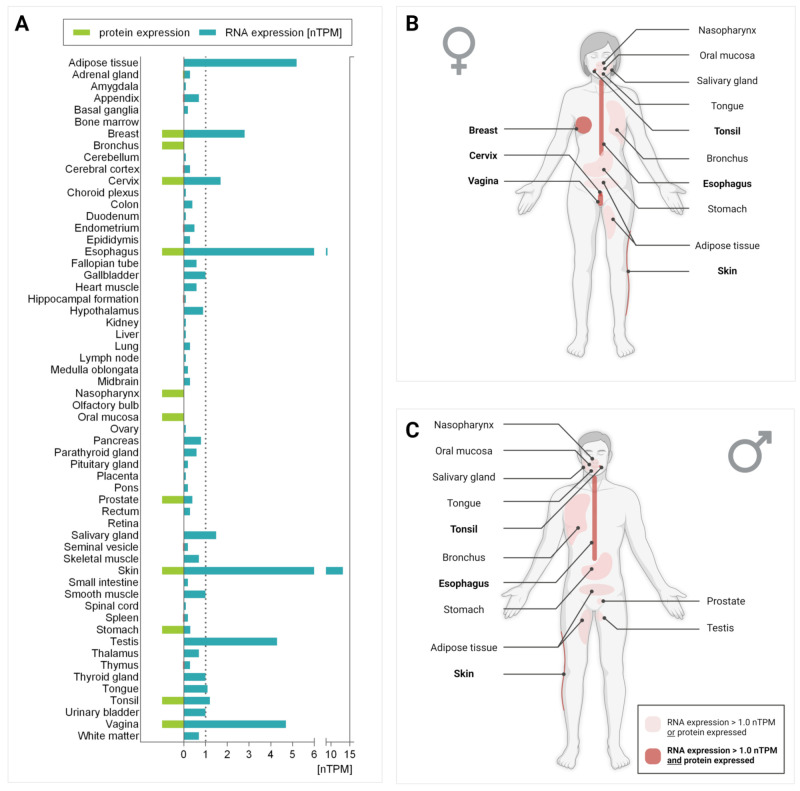
*SLC10A6* mRNA and SOAT protein expression in different human tissues. (**A**) Blue bars indicate *SLC10A6* mRNA expression levels in 58 different human tissues. The raw data were obtained from two different deep-sequencing RNA-seq projects: the Human Protein Atlas (HPA) (www.proteinatlas.org, version V21.0, accessed on 29–31 March 2022) and the Genotype-Tissue Expression (GTEx) project (www.gtexportal.org, accessed on 29–31 March 2022). Values are presented as units of normalized transcripts per million (nTPM) [53]. The absolute numbers of samples per tissue are listed in Appendix A. Values with nTPM < 0.1 were not included. Green bars indicate SOAT protein expression according to the HPA. These data were derived from SOAT antibody-based protein profiling using immunohistochemistry (IHC) (see Figure 6). In tissues where SOAT-specific antibody staining was detectable, a green bar is shown in the graph, regardless of the intensity of the staining. (**B**,**C**) Schematic representation of SOAT expression in the female body and male body, respectively. Tissues where only *SLC10A6* mRNA expression was detected above a cutoff value of 1 nTPM or SOAT protein expression could be detected by IHC are labeled. Tissues in which both *SLC10A6* mRNA (>1 nTPM) and SOAT protein expression were detected are marked in dark red. Figure created with BioRender.com.

**Figure 6 ijms-24-09926-f006:**
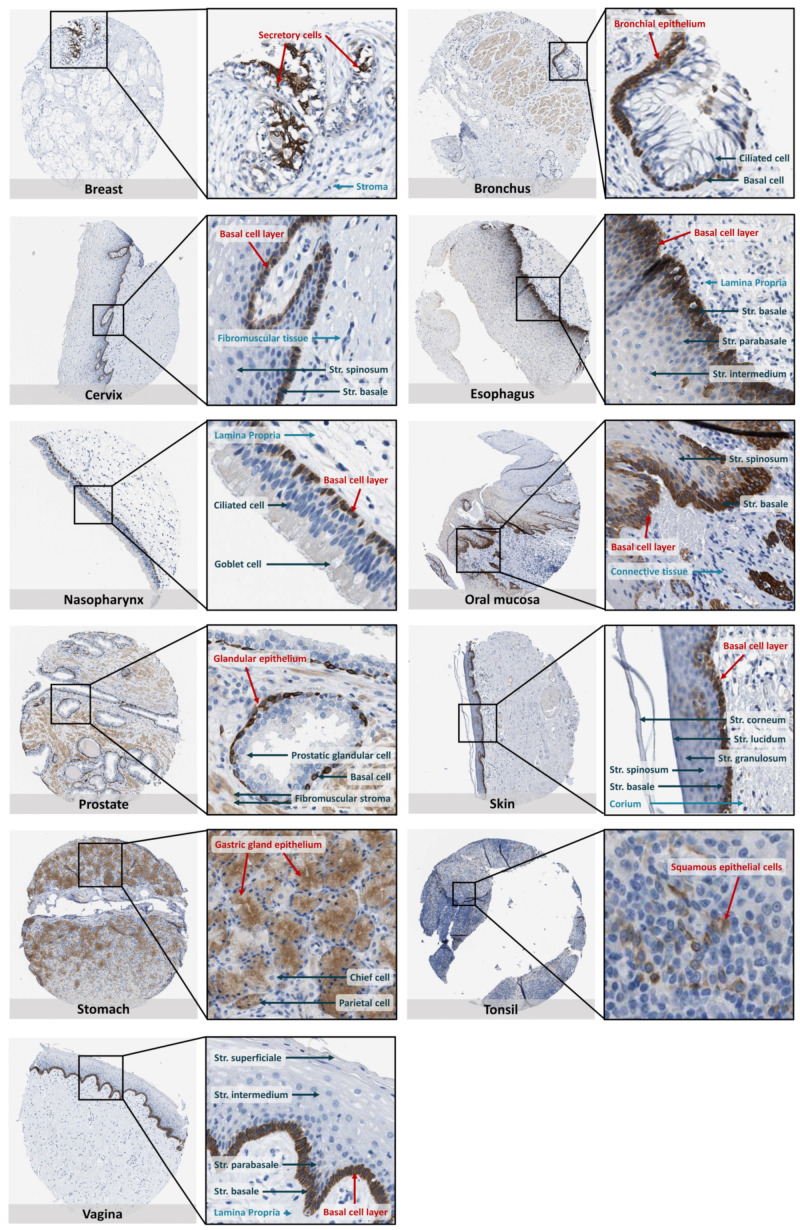
SOAT protein expression in histological sections from non-neoplastic and morphologically normal tissues. All histological images were obtained from the online Human Protein Atlas database (HPA) (www.proteinatlas.org, version V21.0, accessed on 29 March 2022) and represent immunohistochemistry (IHC) detection of the SOAT protein (antibody HPA016662) with 3,3’-diaminobenzidine staining and hematoxylin counterstaining [53,54,55,56]. The URLs of all presented images are listed in Appendix A. In total, 44 different tissues were analyzed. Representative IHC images are shown (an overview and a magnification of each) in which SOAT-staining was cell-type specific. SOAT protein expression is indicated with red arrows. Additionally, typical epithelial and mesenchymal structures of tissues are marked in light and dark blue, respectively.

In addition, individual IHC studies have localized the SOAT protein in germ cells at various stages in human testis biopsies showing normal spermatogenesis [13]. In the human breast, SOAT expression was detected in the ductal epithelium [51], and in the human placenta, SOAT immunoreactivity was detected in the maternal-facing microvillous plasma membrane of syncytiotrophoblasts and vessel endothelium [49]. In contrast, SOAT immunoreactivity was weak in mesenchymal cells and was observed, for example, in the fibromuscular stroma of the prostate (Figure 6). This typical cell-type specific expression pattern of the SOAT was also confirmed by quantitative single-cell transcriptome data from the HPA, which showed dominant SOAT expression in epithelial and endothelial cells of the breast, endometrium, esophagus, prostate, and skin, as well as in early and late spermatids of the testis. In contrast, generally low or absent SOAT expression was detected in mesenchymal cells such as fibroblasts and smooth muscle cells (www.proteinatlas.org, version V21.0, accessed on 29–31 March 2022).

## 11. Other Steroid Sulfate Transporters

As outlined above, the intracrine formation of active steroids via the sulfatase pathway requires the cellular uptake and desulfation of sulfated steroid hormones in peripheral tissues. Therefore, the net import of sulfated steroids can be regarded as a balance between carrier-mediated steroid uptake and efflux [6,57]. Apart from the SOAT, many other uptake and efflux carriers such as DHEAS and E1S have been described that can transport sulfated steroid hormones [52,58]. These include uptake carriers from the organic anion transporting polypeptide (OATP) and organic anion transporter (OAT) carrier families SLCO and SLC22, as well as efflux carriers from the ATP-binding cassette (ABC) family (Table 4). According to the whole-genome transcriptome data from the GTEx project (Figure 7), some of these carriers show a specific expression pattern.

**Figure 7 ijms-24-09926-f007:**
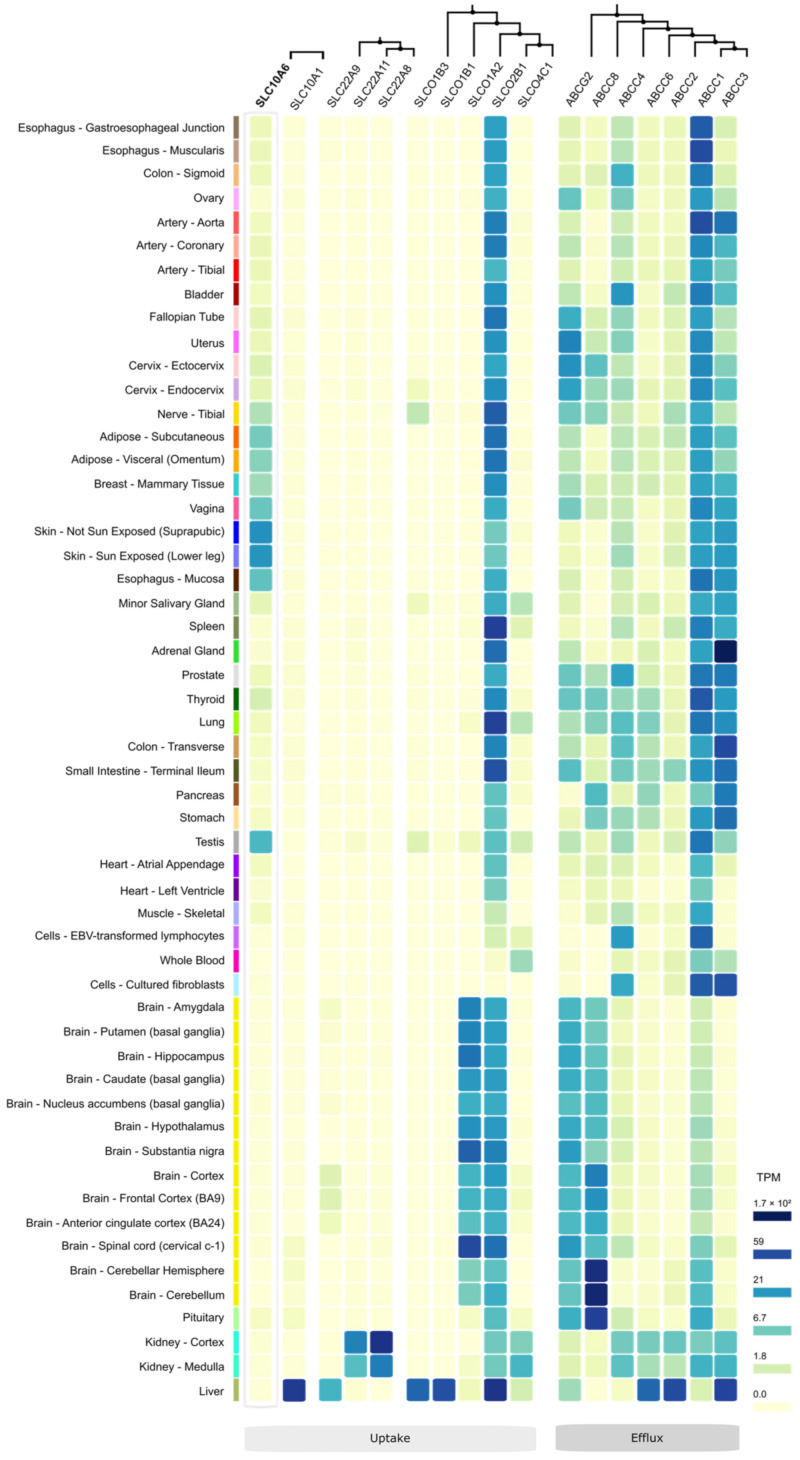
mRNA expression pattern of DHEAS/E1S uptake and efflux transporters. The raw data were obtained from the Genotype-Tissue Expression (GTEx) project (www.gtexportal.org, analysis release V8, dbGaP accession phs000424.v8. p2, accessed on 20 June 2022), via module expression > multi-gene query. Only steroid sulfate carriers were included for which Km values for DHEAS/E1S uptake or efflux are described in the literature (see Table 4). The phylogenetic relationships among the different carrier groups are indicated at the top of the diagram. Color-coding of the tissue expression is based on the mRNA expression level in transcripts per million (TPM) (see legend on the right).

**Table 4 ijms-24-09926-t004:** Uptake carriers of sulfated steroids.

Gene	Protein (Syn.)	DHEAS Transport Km	E1S Transport Km	Other Selected Endogenous Substrates	Substrate Pattern	Transport Mode
*SLC10A1*	NTCP (LBAT)	56.1 µM [14]	57.6 µM [14]	Bile acids, taurolithocholic acid [14]	Multi-specific	Active, Na^+^-dependent
*SLC10A6*	SOAT	28.7 µM [1]	12.0 µM [1]	Testosterone sulfate, taurolithocholic acid [3,14]	Specific	Active, Na^+^-dependent
*SLCO1A2*	OATP1A2 (OATP-A)	7 µM [59,60]	16–59 µM [59,61,62,63]	Bile acids, bilirubin, thyroid hormones, prostaglandin E_2_ [64]	Multi-specific	Na^+^-independent
*SLCO1B1*	OATP1B1 (OATP-C)	22 µM [65,66]	0.09–45 µM [59,65,67,68]	Bile acids, bilirubin, thyroid hormones, glucuronide conjugates, prostaglandin E_2_ [66,69]	Multi-specific	Na^+^-independent
*SLCO1B3*	OATP1B3 (OATP8)	>30 µM [65]	3–58 µM [70,71,72]	Bile acids, bilirubin, thyroid hormones, leukotriene C_4_ [73]	Multi-specific	Na^+^-independent
*SLCO2B1*	OATP2B1 (OATP-B)	9 µM [74]	1.56–21 µM [67,74,75,76]	Taurocholic acid, prostaglandins, leukotriene C_4_, thromboxane B_2_ [77]	Multi-specific	Na^+^-independent
*SLCO4C1*	OATP4C1 (OATP-H)	-	26.6 µM [78]	Bile acids, conjugated steroids, thyroid hormones, eicosanoids [77,79]	Multi-specific	Na^+^-independent
*SLC22A8*	OAT3	13 µM [80]	2.2–21.2 µM [81]	Creatinine, cAMP, glutarate, oxalate, prostaglandins E_2_ and F_2α_ [77]	Multi-specific	Na^+^-independent
*SLC22A11*	OAT4	0.63–29.2 µM [81]	1.01–21.7 µM [81]	Creatinine, glutarate, prostaglandins E_2_ and F_2α_ [77]	Multi-specific	Na^+^-independent
*SLC22A9*	OAT7	2.2 µM [81]	8.7 µM [81]	Creatinine, cGMP [82]	Multi-specific	Na^+^-independent

Note: Only the carriers for which Km values for the transport of DHEAS or E1S are described in the literature are listed.

The carriers NTCP (*SLC10A1*), OAT7 (*SLC22A9*), OATP1B1 (*SLCO1B1*), and OATP1B3 (*SLCO1B3*) are predominantly expressed in the liver. OAT4 (*SLC22A11*), OAT3 (*SLC22A8*), and OATP4C1 (*SLCO4C1*) are mostly found in the kidney, and OATP1A2 (*SLCO1A2*) is typically expressed in the brain. In contrast, OATP2B1 (*SLCO2B1*) is broadly expressed in nearly all human tissues. Regarding steroid sulfate efflux, most of the ABC carriers involved, namely ABCG2, ABCC8, ABCC4, ABCC6, ABCC2, ABCC1, and ABCC3, show broad expression patterns with a preference for the brain (ABCG2, ABCC8), liver (ABCC6, ABCC2), or peripheral organs (ABCC1, ABCC3) (Figure 7). Compared to the steroid sulfate uptake carriers from the OATP and OAT families, the SOAT exhibits some specific features (see Table 4): (1) SOAT transport is active and strictly sodium-dependent, (2) its substrate pattern is very much restricted to sulfated steroid hormones (Table 2), and (3) the SOAT is typically expressed in many steroid-responsive tissues such as the testis, skin, vagina, breast, and adipose tissue (Figure 5, Figure 6 and Figure 7) [52].

## 12. Role of SOAT in Health and Disease

Studies over the past two decades have revealed a potential role of the SOAT in the overall regulation of steroid-responsive tissues and the development of certain diseases, with a particular focus on male fertility/infertility, placenta function during pregnancy, breast carcinogenesis, and chronic inflammation during obesity. In the next section, the key findings of these studies are summarized and discussed in the context of the regulation of SOAT expression in various diseases.

### 12.1. Male Fertility/Infertility

At the mRNA level, the SOAT has shown very high expression levels in the testes of both humans and mice, and its expression has been analyzed more closely at both the mRNA and protein levels in both species. In human testis biopsies showing normal spermatogenesis, SOAT protein expression was detected through IHC in zygotene primary spermatocytes of stage V, pachytene spermatocytes across all stages (I–V), secondary spermatocytes of stage VI, and round spermatids (step 1 and step 2) in stages I and II [13]. This clear germ cell-specific expression pattern of the SOAT in the human testis was further verified through laser-assisted cell picking (LACP) experiments of paraffin-embedded tissues, followed by RT-PCR expression analysis. These experiments only showed SOAT expression in material picked from the seminiferous tubules (containing the peritubular cells, Sertoli cells, and germ cells) but not from the interstitial tissue (encompassing Leydig cells, blood vessels, and connective tissue) [13]. In addition, in mice, the Soat showed a clear germ cell-specific expression pattern, and IHC analysis revealed its localization in primary leptotene and pachytene spermatocytes, as well as in residual bodies of elongating spermatids at stages X and XI of spermatogenesis [4].

Due to this specific expression pattern in the testis, we proposed that the SOAT plays a role in germ cell differentiation toward mature spermatids and, consequently, in male fertility. This was supported by the finding that SOAT expression was significantly lower or even absent in severe disorders of spermatogenesis, characterized by the arrest of spermatocytes or spermatogonia, or the total loss of germ cells (so-called Sertoli cell-only syndrome) [13]. This drop in SOAT expression may play a role in the local supply of sulfated steroid hormones and the overall regulation of steroids during germ cell differentiation and thus may affect male fertility [13]. However, later fertility studies conducted in *Slc10a6* knockout mice (see Section 13) showed that the impairment in Soat-mediated transport of sulfated steroids in the testis did not induce infertility [83].

In humans, a genotype–phenotype correlation study was performed on 20 subjects with normal spermatogenesis and 26 subjects with hypospermatogenesis. All of them were genotyped for the transport of the dysfunctional L204F SOAT polymorphism to compare the allelic frequencies between the two groups [35]. However, both groups showed nearly identical frequencies of the SOAT-L204F polymorphism, with ∼10% heterozygous and ∼5% homozygous subjects, indicating that this particular polymorphism shows no causal relationship with hypospermatogenesis in men. However, lifelong SOAT/Soat knockout, downregulation, or dysfunction might be compensated for by an upregulation of other steroid sulfate carriers, which could conceal the effects and thus make the physiological and pathophysiological role of SOAT/Soat in spermatogenesis unclear (Figure 8, testis).

### 12.2. Placenta Function during Pregnancy

It has been known for a long time that blood concentrations of the estrogens estrone, estradiol, and estriol markedly change in women during pregnancy [84,85]. As an example, maternal serum concentrations of estriol significantly increase in the final months of pregnancy, peaking just before birth [49]. During pregnancy, estriol is produced by the so-called fetoplacental unit. This involves the conversion of DHEAS produced by the fetal adrenals to 16α-DHEAS in the fetal liver, followed by its uptake into placental syncytiotrophoblasts, its desulfation by the steroid sulfatase, and finally, its conversion to estriol, which is then released into the maternal compartment. In addition, DHEAS from the fetal and maternal origin is converted to estradiol [86,87] (Figure 8, placental barrier). As DHEAS and 16α-DHEAS are negatively charged molecules, they need carrier-mediated uptake to enter the syncytiotrophoblast. In this process, SOAT is involved in the uptake of DHEAS via the maternal-facing microvillous plasma membrane of the syncytiotrophoblast and in the transport across the fetal blood vessel endothelium. In addition, the carriers OATP2B1 and OAT4 are involved in the transport of DHEAS and 16α-DHEAS across the basal plasma membrane of the syncytiotrophoblast, which is oriented toward the fetal circulation [49,88,89,90,91]. Low estriol concentrations during pregnancy have been correlated with fetal disorders such as Down syndrome or anencephaly [92,93]. Based on the mechanism for placental estriol synthesis outlined above, genetic defects in one of the DHEAS/16α-DHEAS uptake carriers in the syncytiotrophoblast could be a possible reason for low maternal estriol levels. This should be considered in future studies on maternal estriol levels during pregnancy.

### 12.3. Breast Cancer

Estrogens play a crucial role in the development and proliferation of hormone-dependent breast cancer by interacting with the estrogen receptor (ER) [51,94]. Accordingly, tamoxifen, which blocks the effect of estrogens on ER+ breast cancer cells, has been the cornerstone of endocrine breast cancer therapy for decades [95]. Due to the role of the SOAT in the cellular uptake of sulfated estrogen and androgen precursors such as E1S and DHEAS, this carrier may also be involved in the proliferation of estrogen-dependent breast cancer cells (Figure 8, breast). The SOAT showed relatively high expression levels in breast cancer tissue biopsies, showing no significant association with the tumor grade, stage, receptor status, or age of the patient [51].

In a study investigating the effect of SOAT-mediated E1S uptake into hormone-dependent T47D breast cancer cells, physiologically relevant concentrations of E1S significantly stimulated cell proliferation at high SOAT expression levels. This effect was completely reversed by the SOAT inhibitor 4-SMP. In addition, T47D cells with low SOAT expression levels showed significantly lower proliferation rates [51]. These data led to the conclusion that SOAT-mediated E1S uptake contributes to the overall cell proliferation of hormone-dependent breast cancer cells and that SOAT inhibitors have anti-proliferative potential. Inhibitors of SOAT from different chemical classes have already been identified. Based on the already established structure–activity relationships for some of them, more potent derivatives are expected to be developed in the near future.

### 12.4. Adiposity and Inflammation

Numerous studies have examined the relationship between overweight, central obesity, and plasma levels of DHEA and DHEAS [96,97,98,99,100,101], providing some valuable insights. For example, low serum levels of DHEAS have been associated with a high body mass index, central fat accumulation, and an increase in visceral fat [102,103,104,105]. Conversely, clinical trials have demonstrated that DHEA supplementation can help to decrease the total body fat mass, as well as the abdominal visceral and subcutaneous fat mass [106,107,108,109]. In vitro studies on adipocytes have revealed a stimulatory effect of DHEAS on lipolysis, suggesting that elevated levels of DHEAS may negatively influence lipid accumulation [103]. In addition, obesity has been associated with chronic low-grade inflammation [110]. Interestingly, Kosters et al. (2016) showed that in mice, the *Slc10a6* mRNA expression level was dramatically increased in liver and white adipose tissue after lipopolysaccharide (LPS)-induced inflammation [111]. Based on this effect, an in vitro study with murine 3T3-L1 adipocytes was performed to investigate the effect of Soat expression on adipogenesis during LPS-induced inflammation (Figure 8, adipose tissue) [112]. It was demonstrated that LPS treatment of murine 3T3-L1 adipocytes significantly increased *Slc10a6* mRNA levels. In addition, the combined LPS and DHEAS treatment of cells reduced the accumulation of intracellular lipid droplets and the overall lipid accumulation, most likely via the increased DHEAS uptake after Soat upregulation [112]. Similarly, the LPS treatment of mice led to significantly higher *Slc10a6* mRNA levels in white intra-abdominal, subcutaneous, and perirenal adipose tissue compared to untreated control mice [112]. We, therefore, hypothesize that the overall expression level and import capacity of SOAT/Soat for DHEAS play a regulatory role in lipid accumulation and chronic inflammation in obesity.

### 12.5. SOAT Expression in Cancer Tissues

Based on experimental IHC data and data from the HPA, SOAT expression seems to be upregulated in different cancer types. In the case of breast cancer, strong SOAT immunoreactivity was detected in specimens showing ductal hyperplasia, intraductal papilloma, atypical ductal hyperplasia, intraductal carcinoma, and invasive ductal carcinoma [51]. In addition, strong SOAT immunoreactivity was detected in pancreas carcinoids, cervical cancer, head and neck cancer, lung cancer, skin cancer, and urothelial cancer (Figure 9). In some of these cancer types, the uptake of estrogen and androgen precursors via SOAT may contribute to the proliferation of tumor cells, as demonstrated for hormone-dependent T47D breast cancer cells [51].

To determine if SOAT expression is correlated with the survival probability of patients with specific types of cancer, the expression data from The Cancer Genome Atlas (TCGA) were re-analyzed using Kaplan–Meier analysis. The mean overall survival times were estimated and the differences were measured using the log-rank test, as shown in Figure 10, which shows a comparison of the survival probabilities of patients with either high or low *SLC10A6* mRNA expression levels over time. The cutoff value between the high and low expression rates was defined to result in a maximum difference in survival between the two groups, as determined by the lowest log-rank p-value. As shown in Figure 10, high SOAT expression was significantly associated with a lower probability of overall survival for patients with glioma, breast cancer, renal cancer, urothelial cancer, and endometrial cancer. The mean (95% CI) survival times (days) in these cancer types (high vs. low gene expression) were glioma: 431 (324–537) vs. 588 (456–719); breast cancer: 3918 (3266–4570) vs. 5199 (4557–5841); renal cancer: 2937 (2742–3132) vs. 4475 (4066–4885); urothelial cancer: 1474 (1017–1931) vs. 2419 (2060–2779); and endometrial cancer: 3590 (2433–4747) vs. 4159 (3675–4643). These data support the concept that pharmacological inhibitors of SOAT may be beneficial for the overall survival of tumor patients, at least for the cancer types analyzed in Figure 10.

**Figure 9 ijms-24-09926-f009:**
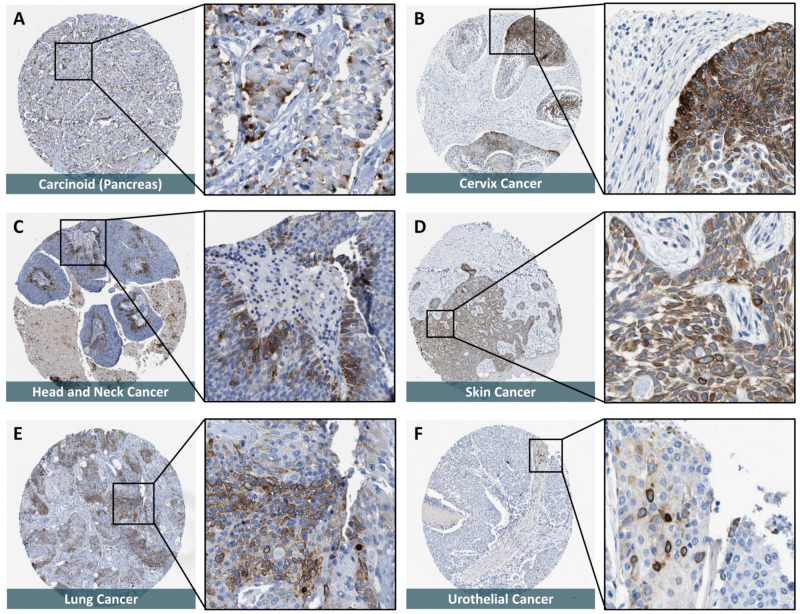
SOAT protein expression in histological sections from cancer tissues. All histological images were obtained from the online Human Protein Atlas (HPA) database (www.proteinatlas.org, version V21.0, accessed on 29 March 2022) and represent immunohistochemistry (IHC) detection of the SOAT protein (antibody HPA016662) with 3,3’-diaminobenzidine staining and hematoxylin counterstaining [53,54,55,56]. The URLs of all presented images are listed in Appendix A. In total, 20 different cancer types from 216 patients were analyzed. Representative IHC images are shown (an overview and a magnification of each), where SOAT-specific staining occurred at a high (**A**–**D**) to moderate (**E**,**F**) intensity. (**A**) Malignant carcinoid of the pancreas. (**B**) Squamous cell carcinoma of the cervix. (**C**) Adenocarcinoma of the salivary gland. (**D**) Basal cell carcinoma of the skin. (**E**) Squamous cell carcinoma of the lung. (**F**) High-grade urothelial carcinoma of the bladder.

**Figure 10 ijms-24-09926-f010:**
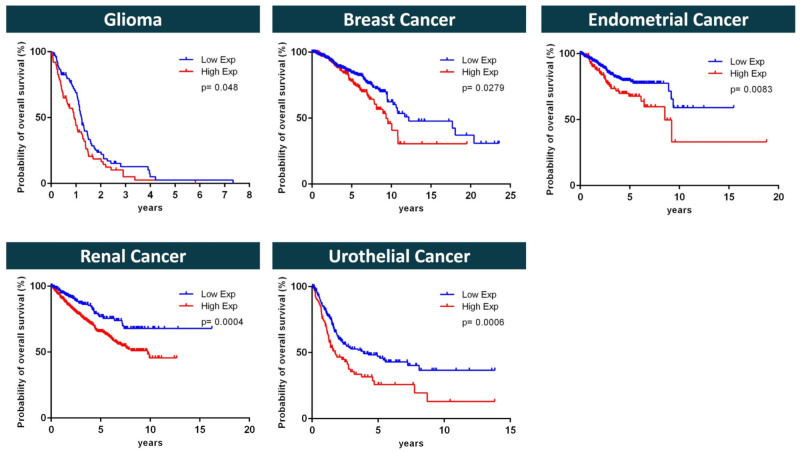
Probability of overall survival of patients with low or high *SLC10A6* mRNA expression, visualized by Kaplan–Meier curves. Data derived from patients with glioma, breast cancer, renal cancer, urothelial cancer, and endometrial cancer. Graphs show the survival probability of patients as a function of time and low (blue curves) or high (red curves) *SLC10A6* mRNA expression (Exp.). The raw *SLC10A6* mRNA expression data were originally derived from The Cancer Genome Atlas (TCGA) and were obtained from the pathology section in the Human Protein Atlas (HPA) (www.proteinatlas.org, version V21.0, accessed on 30 March 2022). The *SLC10A6* mRNA expression levels were analyzed as fragments per kb of exon per million mapped fragments (FPKM) for all available 17 cancer types, namely glioma, thyroid cancer, lung cancer, colorectal cancer, head and neck cancer, stomach cancer, liver cancer, pancreatic cancer, renal cancer, urothelial cancer, prostate cancer, testis cancer, breast cancer, cervical cancer, endometrial cancer, ovarian cancer, and melanoma. Based on the FPKM *SLC10A6* mRNA expression data, patients were divided into two groups (low and high expression), and the relationship between prognosis (survival) and gene expression (FPKM) was examined. For this purpose, the HPA provided a cutoff value that could effectively discriminate between high and low *SLC10A6* mRNA expression. This cutoff value corresponded to the FPKM value, which was chosen to maximize the difference in survival between the two groups with the lowest log-rank p-value. Only the cancer types with a significant difference in the probability of overall survival between patients with low and high *SLC10A6* mRNA expression are shown. Kaplan–Meier curves were used to estimate survival, and the log-rank (Mantel–Cox) test was used to compare the times to events between the low- and high-expression groups.

## 13. *Slc10a6* Knockout Mouse Model

To elucidate the physiological role of the SOAT in the body, we established a *Slc10a6* knockout mouse model with a C57BL/6N genetic background via homologous recombination-based target deletion of coding exons 2 and 3 [83]. Although there were some differences between the quantitative expression levels of the mouse *Slc10a6* and the human *SLC10A6* transcripts, the organ expression patterns with high expression levels in the skin, testis, and lung were generally consistent between both species [1,4]. Due to the high Soat expression in germ cells of the testis, we first analyzed spermatogenesis and performed reproductive phenotyping in these mice. As there were no significant differences in the number of offspring, infantile mortality, vitality of the offspring, gender distribution, or body weight at the time of weaning between *Slc10a6(+/+)* and *Slc10a6(−/−)* mice, the Soat knockout mice were considered to have a normal reproductive phenotype. In addition, there were no significant differences in testis weight, frequency, and duration of spermatogenesis, as well as sperm motility and velocity. However, a more detailed histomorphological analysis revealed that impaired spermatogenesis, total germ cell aplasia, and missing generations of germ cells all occurred more frequently in the *Slc10a6(−/−)* mice, but without reaching statistical significance [83].

As Soat is a specific steroid sulfate transporter in mice as well [4], changes in the serum steroid levels were expected. Thus, the serum levels of the *Slc10a6(+/+)* and *Slc10a6(−/−)* mice were comprehensively analyzed for 12 sulfo-conjugated and 9 unconjugated steroids using LC-MS/MS or GC-MS/MS analysis [83,113]. However, most steroids were either completely absent, detected only in certain samples, or present at concentrations below the limit of quantification. Therefore, statistical data analysis could only be performed for testosterone, corticosterone, and cholesterol sulfate. Interestingly, cholesterol sulfate (CS) was detected in the serum at relatively high concentrations of ~1000–2500 ng/mL and showed significantly higher levels in male *Slc10a6(−/−)* knockout mice compared to wild-type mice. However, this effect was confined to male mice [83]. Of note, there was also a trend toward lower testosterone levels in the serum of the *Slc10a6(−/−)* knockout mice, whereas testosterone was nearly absent from the serum of the female mice.

Since peripheral tissues such as skin and adipose tissue may also be active in intracrine steroid synthesis, we additionally investigated potential differences in the adipose tissue and lipid metabolism between *Slc10a6(+/+)* and *Slc10a6(−/−)* mice [112]. Immunofluorescence studies confirmed the expression of the Soat protein in the plasma membrane of white, subcutaneous, and brown adipocytes, reflecting the high SOAT/Soat mRNA expression levels in these tissues. Hence, the total body weight; mass of white intra-abdominal, subcutaneous, and brown adipose tissue; adipocyte size; and serum levels of the three adipokines, adiponectin, leptin, and resistin, were examined in Soat knockout and wild-type mice. Although there was only a trend for higher adipose tissue weights in the male *Slc10a6(−/−)* mice, significant differences were detected in the adipocyte size. Here, female *Slc10a6(−/−)* mice showed significantly larger adipocytes in white and subcutaneous adipose tissue in contrast to *Slc10a6(+/+)* mice. In addition, male *Slc10a6(−/−)* mice showed significantly larger adipocytes in brown adipose tissue and a trend toward larger adipocytes in white and subcutaneous adipose tissue. All other parameters investigated did not show any significant differences between the *Slc10a6(−/−)* and *Slc10a6(+/+)* mice.

## 14. Cell Models

In addition to the Soat knockout mouse model, cell lines with naturally high SOAT expression are valuable tools for characterizing the *SLC10A6* expression and functional regulation of this membrane carrier. In these cell lines, the inducers and repressors of *SLC10A6* mRNA expression can be investigated, as well as the cellular metabolism of sulfated steroid hormones imported via the SOAT. Systematic genome-wide expression data exist for several human cell lines in the Human Protein Atlas (Figure 11A, Table 5). In the case of the SOAT, several cell lines derived from skin and mesenchymal tissues show relatively high mRNA expression levels >1 nTPM, namely HaCaT, BJ, and BJ hTERT+. In particular, the human skin-derived cell line HaCaT seems to be very promising for SOAT/*SLC10A6* expression studies, as this cell line was originally derived from skin keratinocytes and, therefore, could present an in vitro skin model for SOAT research. In addition, the cell lines ASC diff (from adipose stroma cells), HBEC3-KT (from bronchial epithelial cells), and HeLa (from cervical epithelial cells) are of interest, as they all reflect the in vivo expression pattern of human SOAT.

**Figure 11 ijms-24-09926-f011:**
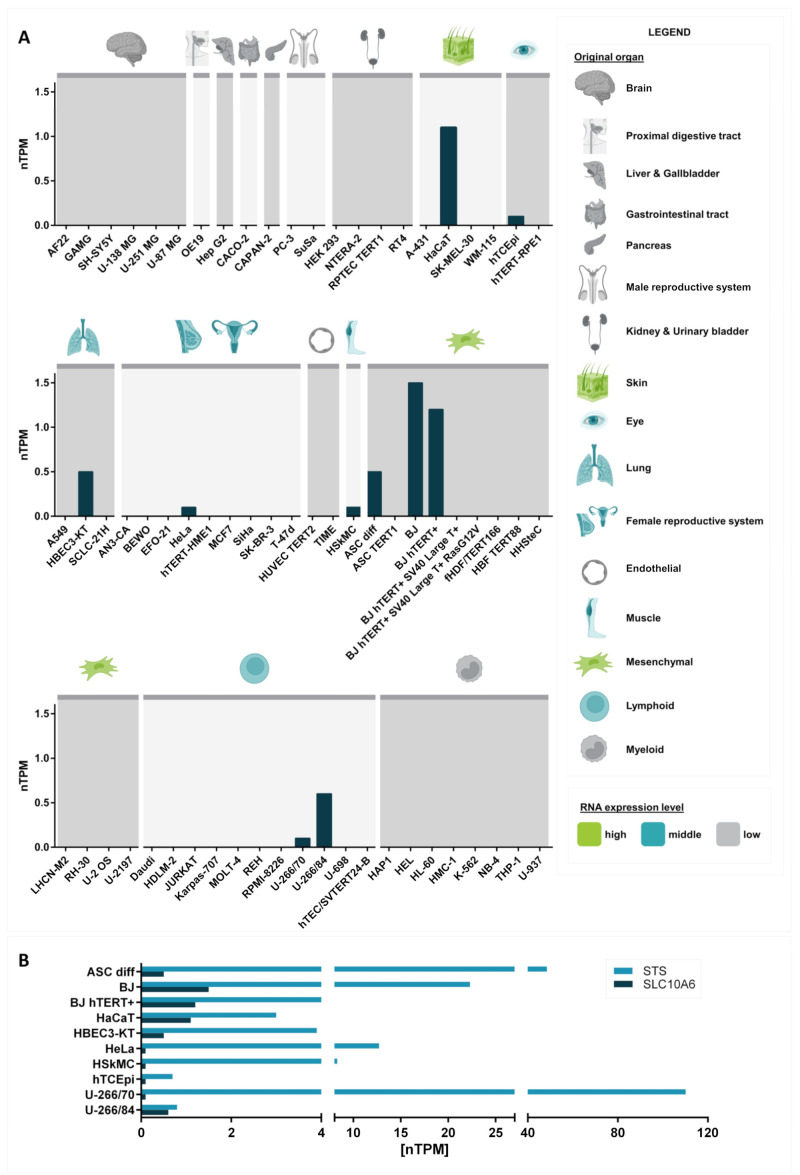
*SLC10A6* and *STS* mRNA expression in different cell lines. The raw data were obtained from the cell line section of the Human Protein Atlas (HPA) (www.proteinatlas.org, version V21.0, accessed on 31 March 2022) that contains expression data of 69 different cell lines. Values are presented as units of normalized transcripts per million (nTPM) [53]. (**A**) Grouping of the cell lines based on the original organ they were obtained from. Organs from which single cell lines showed nTPM values >1 are highlighted in green. Organs from which single cell lines showed intermediate *SLC10A6* expression (nTPM >0 and <1) are labeled in blue. Organs from which none of the established cells showed any SOAT expression are indicated in grey. (**B**) For all cell lines with significant *SLC10A6* mRNA expression (dark-blue bars), expression of *STS* is additionally provided (light-blue bars). Figure created with BioRender.com.

In contrast, the cell lines BJ, BJ hTERT+, HSkMC, hTCEpi, U-266/70, and U-266/84 appear to be less suitable for representative experiments on the SOAT since they do not fully reflect the in vivo tissue- and cell-type-specific *SLC10A6*/SOAT expression pattern. As mentioned above, the SOAT is considered to act in concert with steroid sulfatase to supply active estrogens and androgens to peripheral tissues. Therefore, in addition to the SOAT, studies focusing on the sulfatase pathway should also show sufficient expression of STS in the respective cell line. Data from the HPA show that STS mRNA is expressed in all cell lines that also express *SLC10A6* mRNA, as shown in (Figure 11B). These data underline the suitability of the above-mentioned cell lines as suitable in vitro models for SOAT research.

## 15. Summary and Further Perspectives

Over the last two decades, much has been learned about the expression and function of the SOAT. The structural requirements for SOAT substrates are quite clear and are mostly restricted to 3′- and 17′-monosulfated steroid hormones. Several studies have identified SOAT inhibitors from different chemical classes and a valid pharmacophore model for SOAT inhibitors has been developed. Based on this, more potent and specific SOAT inhibitors can be developed using approaches such as optimized virtual library screening, which allows for the identification of novel inhibitors from other chemical classes.

The role of the SOAT within the concept of intracrine steroid synthesis via the sulfatase pathway is already well-established, although the quantitative contribution of SOAT-mediated steroid sulfate import into specific target cells still requires further investigation. For hormone-dependent breast cancer cells, proof-of-concept SOAT inhibition has already revealed an anti-proliferative effect, inspiring further research on additional cancer cell lines and the development of more potent SOAT inhibitors for anti-proliferative therapy. The development of such pharmacological SOAT inhibitors should benefit from the already established drug developmental programs for the ASBT and NTCP inhibitors. These bile acid transport inhibitors are used, e.g., in the treatment of cholestatic liver disease, against chronic constipation, or as cholesterol-lowering therapy [9]. Indeed, some of the NTCP and ASBT inhibitors have significant cross-reactivity against the SOAT (e.g., betulinic acid, BSP, troglitazone, S 1647, or S 3740).

The Soat knockout mouse model has been established and is available for further research on the significance of the Soat in the context of organs. However, this model has a limitation, that is, DHEAS, which is the predominant sulfated steroid hormone in humans, is almost absent in the mouse. For in vitro studies, several cell lines seem to be suitable for studying the regulation of SOAT expression and the quantitative contribution of, e.g., steroid sulfate transport to the cellular steroid profile and regulation.

## Figures and Tables

**Figure 1 ijms-24-09926-f001:**
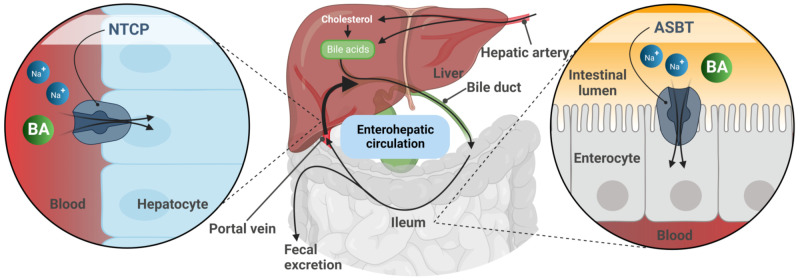
Schematic representation of the enterohepatic circulation of bile acids (BA). After hepatic synthesis and efflux into the bile canaliculi, BA are released into the intestinal lumen and then reabsorbed via the apical sodium-dependent bile acid transporter (ASBT) in the terminal ileum. BA are transported back to the liver with the portal blood flow. Reuptake of BA from portal blood into hepatocytes via the Na^+^/taurocholate co-transporting polypeptide (NTCP) completes the enterohepatic circulation. Figure created with BioRender.com.

**Figure 2 ijms-24-09926-f002:**
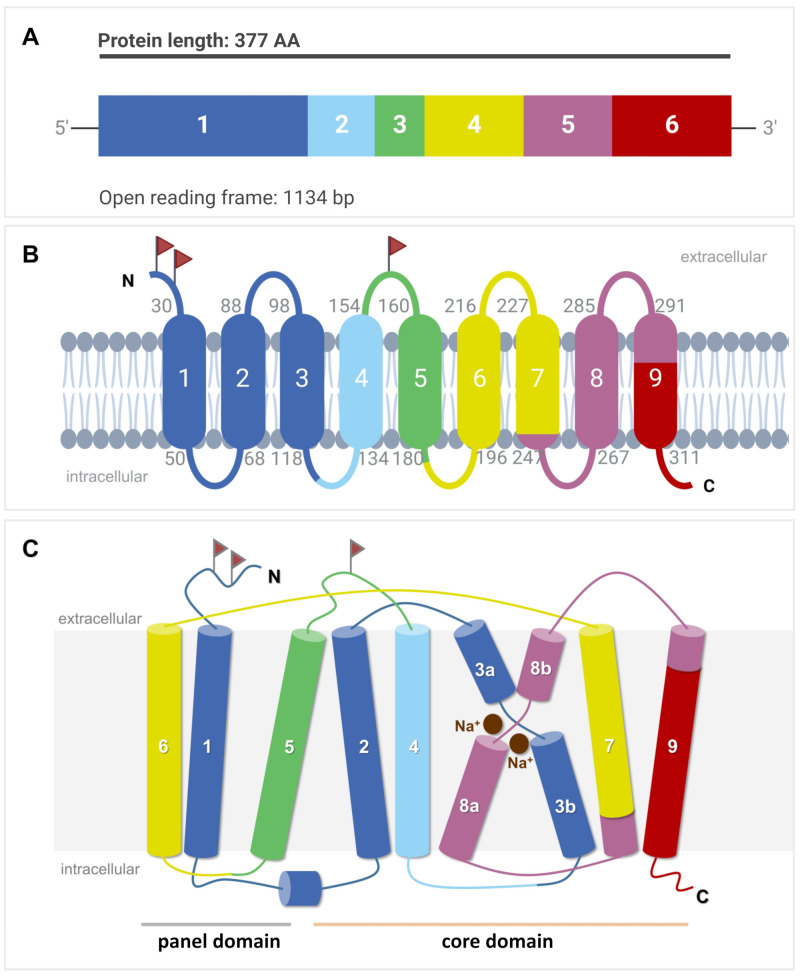
*SLC10A6* mRNA transcript and SOAT membrane topology. (**A**) The open reading frame (ORF) of the *SLC10A6* transcript consists of 1134 base pairs (bp), which derive from six coding exons. This ORF is coding for the 377-amino-acid SOAT protein. (**B**) Schematic transmembrane topology model of the human SOAT protein with nine TMDs and N_exo_/C_cyt_ orientation of the N- and C-terminal ends. The color code indicates which part of the protein is encoded by which corresponding exon. Small gray numbers indicate the amino acid positions of the respective TMD. Red flags indicate possible N-glycosylation sites. (**C**) Structural arrangement of the TMDs in the core and panel domains. TMDs are numbered from 1 to 9. TMDs 3 and 8 are discontinuous and are named 3a/3b and 8a/8b, respectively. Two Na^+^ ions are indicated at the proposed sodium-binding sites. The figure was created with BioRender.com.

**Figure 3 ijms-24-09926-f003:**
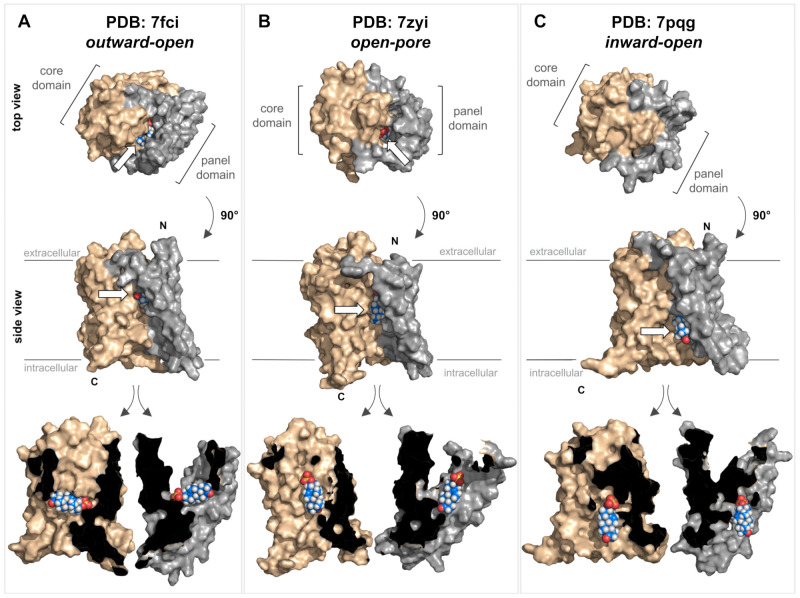
Structure of the SOAT protein and substrate docking. Top and side views of three different homology models of the human SOAT protein (GenBank accession number NP_932069.1) according to SWISS-MODEL predictions (https://swissmodel.expasy.org/ (accessed on 1 March 2023)). The models were generated based on the recent cryo-EM structures of human NTCP and represent three different conformations: outward-open conformation (**A**) PDB: 7fci; open-pore conformation (**B**) PDB: 7zyi; and inward-open conformation (**C**) PDB: 7pqg. In all the models, the core domain is shown in beige and the panel domain is shown in gray. All structures are shown in both top and side views. In addition, the side-view structures were virtually cut between the core and panel domains, and each was turned outwards to visualize the core/panel interface. All three homology models of human SOAT were used for in silico docking of the SOAT substrate DHEAS (indicated by white arrows). Briefly, the identification of potential binding sites was conducted using the Schrödinger SiteMap program [42] and the receptor grid was set to the interface between the core and the panel domain. The DHEAS structure (downloaded from PubChem at pubchem.ncbi.nlm.nih.gov) was prepared for docking using the Schrödinger LigPrep program [43]. Afterward, the DHEAS molecule was docked to the SOAT protein using the Schrödinger Glide XP program [44,45,46] with default settings. All the structures were visualized with PyMOL (version 2.5.4) [47]. N, N-terminus; C, C-terminus.

**Figure 4 ijms-24-09926-f004:**
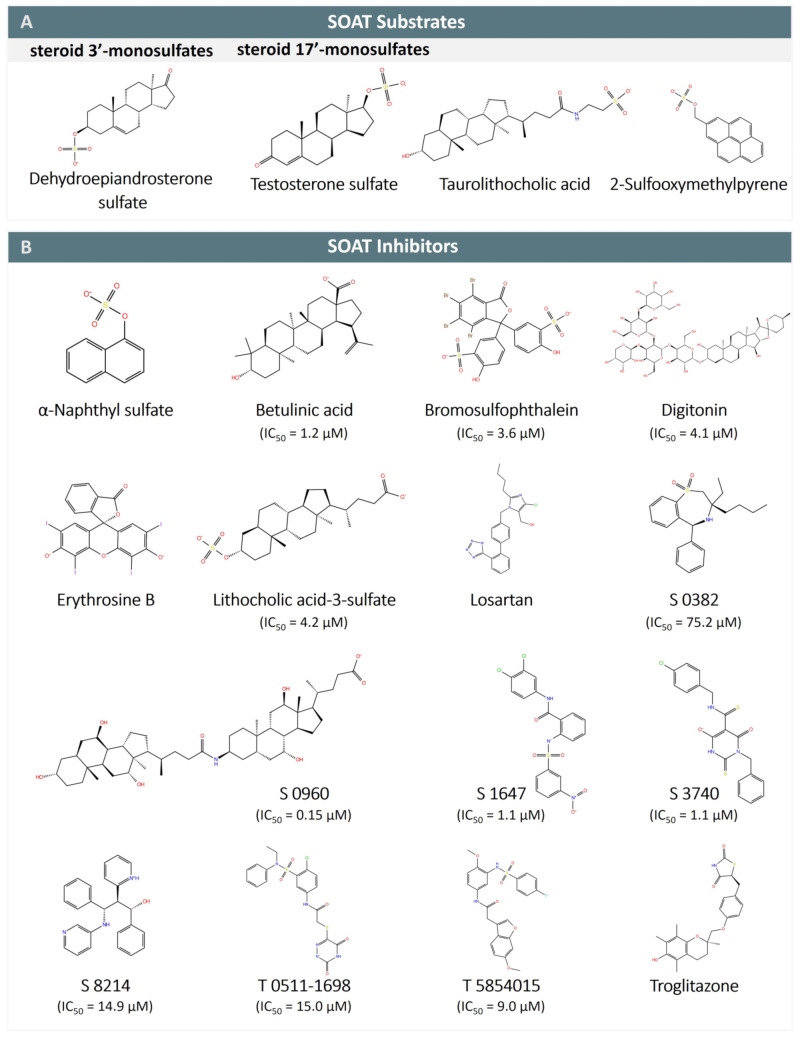
SOAT substrates and inhibitors. (**A**) SOAT specifically transports steroid 3′- and 17′-monosulfates, represented here by dehydroepiandrosterone sulfate (DHEAS) and testosterone sulfate, respectively. Taurolithocholic acid (TLC) is the only non-sulfated bile acid that is transported by SOAT. 2-sulfooxymethylpyrene (2-SMP) and 4-sulfooxymtehylpyrene (4-SMP) are the only non-steroidal SOAT substrates identified so far. (**B**) SOAT inhibitors belong to different chemical classes, including steroid-based compounds (bile acids and heart glycosides), organosulfates (α-naphthylsulfate, bromosulfophthalein BSP), phenyl sulfonamides (S 1647, T 0511-1698, and T 5854015), benzothiazepines (S 0382), barbiturates (S 3740), propanolamines (S 8214), thiazolidinediones (troglitazone), and some others. Some of the SOAT inhibitors are also inhibitors of NTCP (e.g., betulinic acid, BSP, erythrosine B, losartan, and troglitazone) and ASBT (e.g., S 0382, S 0960, S 1647, S 3740, S 8214, and troglitazone).

**Figure 8 ijms-24-09926-f008:**
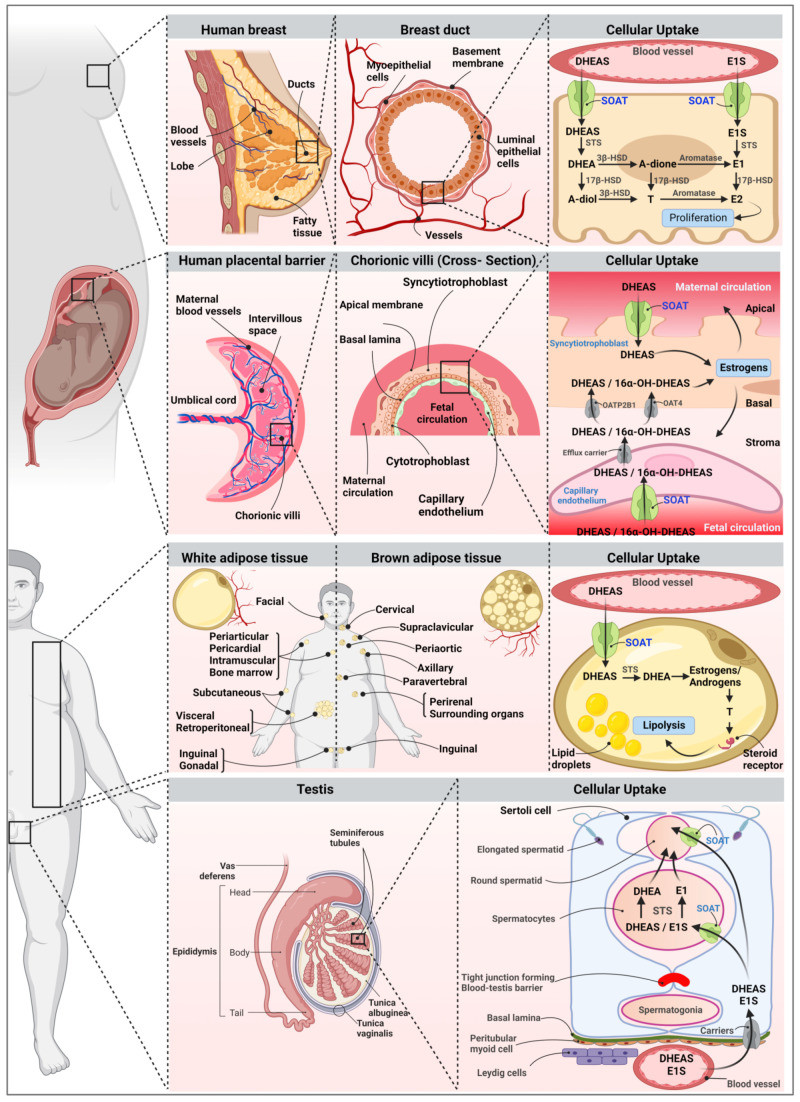
Role of SOAT in supplying sulfated steroid hormones to peripheral tissues. After the uptake of DHEAS and E1S via SOAT, these sulfate steroids can be converted to active steroids by steroid sulfatase (STS). Local steroid production via the sulfatase pathway then contributes to the overall estrogen and androgen regulation of the respective cell types and organs. Schematic illustration of SOAT expression in ductal epithelial cells in the breast, fetal capillary endothelium and apical membrane of syncytiotrophoblasts in the placenta, plasma membrane of adipocytes in adipose tissue, and spermatocytes and spermatids in the seminiferous tubules of the testes. Figure created with BioRender.com.

**Table 1 ijms-24-09926-t001:** Phylogenic relationship of the SLC10 carriers and their substrate patterns.

	Gene	Chromosome	Protein	Protein Length	Transport Substrate		
					Steroid Sulfates	Bile Acids	TLC
** 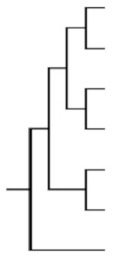 **	*SLC10A1*	14q24	NTCP	349 aa	+	+	+
	*SLC10A4*	4p11	P4	437 aa	−	−	−
	*SLC10A2*	13q33	ASBT	348 aa	−	+	+
	*SLC10A6*	4q21	SOAT	377 aa	+	−	+
	*SLC10A5*	8q21	P5	438 aa	−	−	−
	*SLC10A3*	Xq28	P3	477 aa	−	−	−
	*SLC10A7*	4q31	RCAS	340 aa	−	−	−

Note: “+”, substrate of this carrier; “−“, no substrate of this carrier; aa, amino acids. The phylogenetic relationships among the carriers are indicated on the left.

**Table 5 ijms-24-09926-t005:** *SLC10A6*-expressing cell lines.

Cell Line	Organ	Origin	Cell Type	Category
ASC diff	Mesenchymal	Adipose tissue	Adipose stromal cells	Uncategorized
BJ	Mesenchymal	Foreskin	Fibroblast	Finite
BJ hTERT+	Mesenchymal	Foreskin	Fibroblast	Telomerase immortalized
HaCaT	Skin	Skin	Keratinocyte	Spontaneously immortalized
HBEC3-KT	Lung	Central lung bronchiole	Bronchial epithelial cell	Telomerase immortalized
HeLa	Female reproductive system	Cervix	Epithelial cells derived from cervical cancer cells	Cancer
HSkMC	Muscle	Trapezius and erector spinae muscles	Skeletal muscle cells	Uncategorized
hTCEpi	Eye	Cornea	Corneal epithelial cells	Telomerase immortalized
U-266/70	Lymphoid	Peripheral blood	Cells derived from multiple myeloma cells	Cancer
U-266/84	Lymphoid	Peripheral blood	Cells derived from multiple myeloma cells	Cancer

## Data Availability

Publicly available datasets were used in this study. These data can be found at www.proteinatlas.org (accessed on 29–31 March 2022) (keyword: SLC10A6) and www.gtexportal.org (accessed on 29–31 March 2022 and 20 June 2022) (keyword: SLC10A6).

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
