# Peer review of "Role of the Sodium-Dependent Organic Anion Transporter (SOAT/SLC10A6) in Physiology and Pathophysiology"

_ijms, 2023, doi:10.3390/ijms24129926_

Round 1

Reviewer 1 Report

This review comprehensively summarizes experimental studies and public (genomic/transcriptomic/proteomic) databases on the sodium-dependent organic anion transporter (SOAT/SLC10A6), which specifically transports 3’- and 17’-monosulfated steroid hormones into specific target cells. It also provides insightful perspectives for future exploration of SOAT as a drug target for steroid-responsive human diseases. The manuscript is well-written and organized.   

This reviewer recommends accepting this manuscript for publication after correcting the following typo errors.

1)      Line 99: “no substrate of this inhibitor” should be “not substrate of this carrier”.

2)      Line 573: “I the case of breast cancer” should be “In the case of breast cancer”.  

Author Response

We thank the reviewer for this positive evaluation of our article. We have made the two mentioned typo corrections.

Reviewer 2 Report

The review is well organized and very interesting. Despite almost 20-years history of SLC10 family transporters studies, further studies are still needed to finally establish the sodium-dependent organic anion transporter SOAT as a potential target for new drugs. I can state that the topic is described in appropriate way and the authors performed very detailed literature review of SOAT role in human organism. However, in reviewed manuscript I found critical remarks related to computational part of review and described docking studies (ligand binding points to SOAT). I believe after some simple changes this paper is viable for publication and I will list my comment below.

In figure 2 could be observed SOAT protein structure with docked DHEAS. Docking experiment methods are described in figure caption. I guess that experiment was conducted by authors but that's not what the text says. The reader of the text may feel unsatisfied because of the very laconic description. In reference [14] (Grosser et al) I found that authors have experience in molecular docking. I recommend to move docking description to new chapter with appropriate citations or at least perform richer description in supplementary materials.

For this reason, I recommend publication after major revision.

Author Response

We thank the reviewer for this valuable comments. We agree with the reviewer that the substrate docking part was not sufficiently presented. We have intensely reworked this part. We separated Figure 2 into new Figure 2 and Figure 3 for better clarity. We improved the visualization of the docking data and added a completely new section "5. SOAT substrate docking and proposed transport mechanism" (highlighted in red in the revised manuscript), where we describe and discuss the proposed transport mechanisms of SOAT and the substrate docking data. 

We are convinced that we could significanlty improve this part of the manuscript and hope this can now be accepted for publication.

Round 2

Reviewer 2 Report

Authors addressed all the comments raised by the reviewer. The computational part of review is currently very well described. Now the manuscript can be accepted for publication.

Author Response

Dear reviewer,

thank you very much for your time and the positive evaluation of our manuscript.

Best regards,

Joachim Geyer